# Factors Affecting Enteric Emission Methane and Predictive Models for Dairy Cows

**DOI:** 10.3390/ani13111857

**Published:** 2023-06-02

**Authors:** Andrea Beltrani Donadia, Rodrigo Nazaré Santos Torres, Henrique Melo da Silva, Suziane Rodrigues Soares, Aaron Kinyu Hoshide, André Soares de Oliveira

**Affiliations:** 1Dairy Cattle Research Laboratory, Universidade Federal de Mato Grosso, Campus Sinop, Sinop 78555-267, MG, Brazil; andreadonadia@hotmail.com (A.B.D.);; 2College of Natural Sciences, Forestry, and Agriculture, The University of Maine, Orono, ME 04469-5782, USA; aaron.hoshide@maine.edu; 3AgriSciences, Universidade Federal de Mato Grosso, Campus Sinop, Sinop 78555-267, MG, Brazil

**Keywords:** dairy cattle, feed efficiency, greenhouse gas emissions, modeling

## Abstract

**Simple Summary:**

Meeting the growing animal food demand while reducing greenhouse gas emissions (GGEs) remains one of the challenges for global livestock industries. Enteric methane emission (EME) is the main GGE source in dairy cattle systems. Therefore, evaluating the drivers of EME intensity (per animal product) and developing more accurate predictive models of EME may contribute to dairy cattle sustainability. In this study, we built a large and intercontinental experimental dataset to: (1) explain the effect of EME yield (g methane/kg diet intake) and feed conversion (kg diet intake/kg milk yield) on EME intensity (g methane/kg milk yield); (2) develop models from predicting EME (g/cow/day); and (3) compare the proposed models with 43 external models. Increasing the milk yield reduced EME intensity, and this effect was more due to the enhancement in feed conversion than EME yield. Our models predicted methane emissions better than most external models, with the exception of only two other models which had similar adequacy. Our findings confirm that the improvement in feed conversion should be prioritized for reducing methane emissions in dairy cattle systems.

**Abstract:**

Enteric methane emission is the main source of greenhouse gas contribution from dairy cattle. Therefore, it is essential to evaluate drivers and develop more accurate predictive models for such emissions. In this study, we built a large and intercontinental experimental dataset to: (1) explain the effect of enteric methane emission yield (g methane/kg diet intake) and feed conversion (kg diet intake/kg milk yield) on enteric methane emission intensity (g methane/kg milk yield); (2) develop six models for predicting enteric methane emissions (g/cow/day) using animal, diet, and dry matter intake as inputs; and to (3) compare these 6 models with 43 models from the literature. Feed conversion contributed more to enteric methane emission (EME) intensity than EME yield. Increasing the milk yield reduced EME intensity, due more to feed conversion enhancement rather than EME yield. Our models predicted methane emissions better than most external models, with the exception of only two other models which had similar adequacy. Improved productivity of dairy cows reduces emission intensity by enhancing feed conversion. Improvement in feed conversion should be prioritized for reducing methane emissions in dairy cattle systems.

## 1. Introduction

Meeting the growing demand for animal food while reducing greenhouse gas emissions remains one of the main challenges of the livestock industry. Enteric methane in ruminants is a product mainly of carbohydrate ruminal degradation, and it is eliminated through eructation and respiration [1]. Enteric methane emissions (EMEs) represent a source of gross energy loss ingested by the animal (2 to 12%), where these losses are related to animal feed intake, diet composition, rumen microbial population, physiology, and genetics, as well as the use of dietary additives such as methane inhibitors [1,2,3]. In addition, EME is the major greenhouse gas emissions source in dairy cattle systems [4,5]. Therefore, evaluating the drivers of EME intensity (methane per milk yield) and developing more accurate and user-friendly predictive models of EME per animal may contribute to improving the sustainability of managing dairy cattle. 

Considerable advances have been made in dairy cattle nutrition, genetics, management, and health over the past century, which have contributed to improving dairy cattle productivity and reducing EME intensity mainly in developed countries [6]. An international collective effort to reduce global methane emissions by at least 30% from 2020 levels by 2030 was launched at COP 26 with the Global Methane Pledge [7]. Several opportunities exist to reduce enteric methane, especially in developing countries with large dairy cattle populations and overall middle–low dairy productivity. Reduction in EME intensity from milk yield improvement has been widely reported [4,8]. Dilution of the nutrient requirements for maintenance [9] has been considered the main driver explaining the reduction in EME intensity from animal productivity improvement [6]. We propose here that the EME intensity (g methane/kg milk yield (MY)) may be explained and decomposed into two components. The first component is EME yield (g methane/kg dry matter intake (DMI)), while the second component is feed conversion (kg DMI/kg MY). Our first hypothesis is that feed conversion has a greater impact on EME intensity than EME yield.

Several empirical models to predict EME per animal in dairy cattle have been developed since the 1930s [10,11,12,13,14]. These models have been based mainly on intake and diet characteristics such as gross energy, digestible energy, fiber, and fatty acids. However, as diet characteristics are not routinely measured in several producers, predicting EME at the farm level may be limited. Intake level has been widely reported as the main driver of EME in ruminants [10,11,12,13,14]. Since body weight (BW) and MY have a high correlation with DMI [15], our second hypothesis is that EME models from only animal inputs (BW and MY) may have suitable prediction capabilities. Not rejecting this hypothesis could facilitate both EME prediction and environmental analysis for dairy cattle farms. In addition, as the application of empirical models is population-dependent, accuracy and precision depend also on the dataset size and amplitude. Therefore, our third hypothesis is that new empirical models from a wide intercontinental experimental dataset can improve the prediction of EME from dairy cows. 

The goal of this research is to determine what animal and diet characteristics influence EME in dairy cattle in order to develop predictive models and identify the best strategies to reduce this type of greenhouse gas emission. In this study, we built a large and intercontinental experimental dataset from previous research to: (1) explain the effect of EME yield and feed conversion on EME intensity; (2) develop models for predicting EME (g/cow/day) from dairy cows; and (3) to compare the models developed with several external models that have already been published in the literature.

## 2. Materials and Methods

### 2.1. Dataset

A large and intercontinental dataset was built by a search of peer-reviewed papers using the terms “dairy cows” and “methane” in the Science Direct and Web of Science databases. A total of 225 peer-reviewed papers were initially found for articles using the keywords “dairy cows” and “methane”. To manage the articles, EndNote Web^®^ (Clarivate Analytics) was used. The inclusion criteria of studies were: (1) experiment conducted with lactating cows; (2) presentation of the mean value of the EME per treatment and respective measure of precision of the mean estimate, such as standard error of the mean (SEM), standard deviation, coefficient of variation, or standard error of the difference (SED). When SED was reported in studies evaluated as fixed models, SEM was calculated as SEM = SED/√2. A flowchart that describes the criteria for identifying, excluding, and including articles is available in Appendix A. 

Based on these established inclusion criteria, 115 peer-reviewed papers with 125 experiments and 419 treatment means containing EME were selected (Table 1). The complete dataset as an Excel^®^ file is available in an open research data repository [16], and references of studies are available in Appendix A. The complete dataset was used for comparison of methods to quantify EME and for evaluation of EME intensity (decomposition into EME yield and feed conversion). 

For developing and evaluating predictive models of EME (g/cow/day), the 115 peer-reviewed papers were randomly divided into two subsets. The first subset was used to develop EME models, containing 80 peer-reviewed papers (90 experiments with *n* = 301 treatment means of EME; Table 2). The second subset was used to evaluate the prediction and accuracy of the models, containing 35 peer-reviewed papers (35 experiments with *n* = 118 treatment means of EME Table 3). This procedure was adopted, thus ensuring independence of the dataset used in model development and evaluation.

### 2.2. Comparison of Methods to Quantify Enteric Methane Emission

As our dataset contains studies with different methods for measuring EME, a comparison of animal performance, diet, and EME from studies that used respiration chamber, sulfur hexafluoride (SF6), Greenfeed^®^, and other methods was conducted. As the respiration chamber is considered standard [17], we only included EME data from SF6 and Greenfeed^®^ if EME yield and EME intensity values between studies with different methods were similar to results when using respiration chamber.

A meta-analysis was performed using a mixed linear model, animal variables, diet composition, and EME (g/cow/day; g/kg DMI; and g/kg milk) as dependent variables, methods for EME measuring as a fixed effect, and experiment as a random effect [18]:(1)Y=Xβ+Zγ+ε,
where *Y* represents the vector of dependent variables (treatment means), *β* is an unknown vector of the fixed effect parameters of methods for EME measuring (studies) associated with the known matrix *X*, *γ* is the unknown vector of the study random effect parameter associated with known matrix *Z*, and *ε* represents the vector of the random error with normal distribution (0, σ^2^). Each treatment mean was weighted using the inverse of the SEM of EME (g/cow/day).

### 2.3. Decomposition of Enteric Methane Emission Intensity

We proposed to explain EME intensity from two components as:EME intensity (g methane/kg MY) = EME yield × Feed conversion(2)
where EMEs yield (g methane/kg DMI) and feed conversion. To investigate the influence of these two components on EME intensity, a variance component analysis was conducted [18] considering study as a random effect and EME yield and feed conversion as fixed effects, according to model: (3)Y=Xβ+Zγ+ε,
where *Y* represents vector of the observed EME intensity, *β* is the unknown vector of the fixed effect parameters of the EME yield and feed conversion associated with the known matrix *X*, *γ* is the unknown vector of the study random effect parameter associated with the known matrix *Z*, and *ε* represents the vector of random error with normal distribution (0, σ^2^). In addition, the relationship between MY and EME intensity and yield, as well as feed conversion, was evaluated using a linear mixed model with a variance component structure [19] or a nonlinear mixed model [18]. Each treatment mean was weighted using the inverse of the SEM of EME (g/cow/day). The critical level of probability for type I error was set at α = 0.05. Data points (outliers) were removed in the final models if the studentized residual was outside the range of −2.5 to 2.5 standard deviation from the mean.

### 2.4. Model Development for Predicting Enteric Methane Emission

We developed six models to predict EME measured in g/cow/day. These six models were divided into three groups for evaluating the isolated or combined effects of animal and diets as inputs, with DMI: (1) Animal, which included only animal as predictive variable, without (Animal model I) and with DMI (Animal model II); (2) Diet, which included only diet as predictive variable, without (Diet model I) and with DMI (Diet model II); and (3) Animal + Diet, which included both animal and diet as predictive variables, without (Animal + Diet model I) and with DMI (Animal + Diet model II). 

Animal and diet variables were initially selected from Pearson correlation with EME (g/cow/day) using the complete dataset (Table 1). Animal and diet predictor variables with high correlation with each other (r > 0.70) were not used together in the model to avoid effects from multicollinearity (except for DMI). Multicollinearity occurs when there is a high degree of correlation between independent variables, which reduces the validity of parameter estimates from statistical regression models. Variables with the highest correlation with EME were chosen.

The regression equations were adjusted using mixed linear models, including study as a random effect and the factors associated with the animal, diet, and DMI as fixed effects, according to the following general model [18]:(4)Y=Xβ+Zγ+ε,
where *Y* is the vector of observed EME (g/cow/day), *β* is an unknown vector of the fixed effect parameters of the animal, diet, and DMI associated with the known matrix *X*, *γ* is the unknown vector of the study random effect parameter associated with the known matrix *Z*, and *ε* represents the vector of the random error with a normal distribution (0, σ^2^). Each treatment mean was weighted using the inverse of the SEM of EME (g/cow/day). In addition, variance components were analyzed for each mixed model [18] to identify the main drivers of EME. The critical level of probability for type I error was set at α = 0.05. Outliers were removed in the complete dataset if the studentized residual was outside the range of −2.5 to 2.5 standard deviation from the mean. A list of removed observations is available in Donadia and Oliveira, 2023 [16].

### 2.5. Model Evaluation for Predicting Enteric Methane Emission

The adequacy of the six proposed EME predictive models in the present study was assessed for precision and accuracy by linear regression analysis of the observed values (*Y*) with those predicted (*X*), using the following procedures: graphical analysis, coefficient of determination, mean square of prediction error (MSPE), MSPE decomposition into three sources of error (error due to bias, error due to slope of regression between observed and predicted values to be different from 1, and random error; [20]), concordance correlation coefficient (CCC) and its decomposition into precision (r, correlation coefficient), and accuracy (C_b_, bias correction factor) indicators [21]. 

In addition, the proposed models were compared with 43 currently external models from the literature (Appendix A) from root MSPE (RMSPE; % observed EME) and its confidence interval with type I error of 5% using SEM [22]. Observations with RMSPE studentized residue outside the range of −1.5 to 1.5 standard deviation were considered outliers and removed from model evaluations [22]. In case of prediction error due to bias (intercept of regression between observed versus predicted value different from zero), the intercepts of the proposed models were re-parameterized to annul the prediction error due to bias [22].

## 3. Results

### 3.1. Dataset

The dataset’s studies from the literature measuring enteric methane emissions (EMEs) in dairy cattle were from 18 countries and showed a wide variation in animal characteristics, diet, and methane emissions (Table 1). The USA was the main country where studies were conducted (17.6% of data), followed by Canada (13.6%). The method for quantifying EMEs by respiration chamber represented 51.8% of the data, followed by SF6 (33.7%), and Greenfeed^®^ (12.6%). Rotative experimental designs were adopted by most studies (58.1%), and Holstein was the main genetic group of cows (85.4% of cows). The total mixed ration was the main feeding system (79.0%). Silage and hay represented the main forage sources, 59.1% and 16.2% of the observations, respectively. Subsets used for model development (Table 2) and evaluation (Table 3) presented similar descriptive statistics, indicating that the two subsets were adequately randomized.

### 3.2. Methods to Quantify Enteric Methane Emission

The mean value of EME production (cows/day) from studies using a respiration chamber was higher (*p* < 0.01) than studies with SF6, but it was similar (*p* > 0.05) to those using GreeFeed^®^ and other methods (Table 4). The standard error of mean of the EME production (cow/day) from studies using a respiration chamber was lower (*p* ≤ 0.05) than studies with SF6 and similar (*p* > 0.05) to those using GreenFeed^®^ and other methods (Table 4). However, the differences in means and SEM of the EME may have been due to differences in DMI and MY between studies (Table 4). Thereby, when EMEs were evaluated as yield (g methane/kg DMI) or intensity (g methane/kg MY), no differences were observed (*p* > 0.05) between studies with the different methods (Table 4). Therefore, all studies (using any of these methods) were used this analysis.

### 3.3. Decomposition of Enteric Methane Emission Intensity

We decomposed the EME intensity (g methane/kg MY) in two components: (1) EME yield (g methane/kg MY) and (2) feed conversion (kg DMI/kg MY); EME intensity = EME yield × feed conversion. Feed conversion had a greater contribution in the variation of EME intensity (59.0%) than EME yield (39.6%) (Figure 1a). Increasing the milk yield affected the EME intensity (*p* < 0.01) in a quadratic relationship, with an estimated minimum value of 9.8 g EME/kg milk with 42.7 kg MY/cow/day (Figure 1b). We observed that EME yield had poor goodness of fit with MY (R^2^ = 0.037; Figure 1c). However, feed conversion had an adequate goodness of fit with MY (R^2^ = 0.74; Figure 1d) and with similar behavior to EME intensity (Figure 1b).

Based on our equations, an MY increase of 10 to 40 kg/cow/day reduces EME intensity by 57.7% (23.78 to 10.05 g methane/kg MY; Figure 1b), EME yield by only 9.6% (21.48 to 19.41 g methane/kg DMI; Figure 1c), and feed conversion by 60.0% (1.35 to 0.54 kg DMI/kg MY; Figure 1d). Therefore, the reduction in EME intensity with MY increase was more due to the enhancement in feed conversion than EME yield, indicating that feed conversion (the inverse of feed efficiency) is the main driver responsible for reducing EME intensity in dairy cows.

### 3.4. Proposed Models for Predicting Enteric Methane Emission

Animal inputs with the highest correlation with EME (g/cow/day) were metabolic weight (BW^0.75^) and MY, which were then selected for the Animal models (Table 5). Ether extract (EE) and organic matter digestibility (OMD) were the diet inputs selected for the Diet models (Table 5). Dry matter intake was the variable with the highest correlation with EME (g/cow/day; Table 5). Breed and feeding system were not included as fixed effects in models because our dataset is formed predominantly by Holstein cows feeding with total mixed ration.

The six proposed prediction models of EME (g/cow/day) are described in Table 6. The Animal II, Diet II, Animal + Diet I, and Animal + Diet II models initially had significant (*p* < 0.05) prediction errors due to mean bias (intercept of the linear regression between observed and predicted EME different from zero). Thus, the intercepts of these models were re-parameterized to annul the prediction error due to bias. 

Animal I (without DMI) and II (with DMI) models were adjusted (*p* < 0.05) with MY and BW^0.75^ as predictor variables. Although BW^0.75^ and MY had similar correlations with EME (r = 0.358 and r = 0.385; Table 5), MY accounted for EME total variance more than BW^0.75^ in these two models (Figure 2). The inclusion of DMI as a predictive variable increased the variance due to the fixed effect (r^2^) in the Animal model by ~14% units (Animal model I r^2^ = 24.31%; Animal model II r^2^ = 38.28%; Figure 2). 

Diet models were adjusted with EE and OMD as predictor variables. Inclusion of DMI as a predictive variable increased r^2^ from 1.68% (Diet model I) to 49.7% (Diet model II) (Figure 2). Therefore, OMD intake was the main input of Diet model II (43.62% of EME total variance; Figure 2). 

Animal + Diet models I and II were adjusted (*p* < 0.05) with all inputs used in Animal and Diet models. Animal + Diet model I (without DMI) accounted for only 42.36% of the observed EME total variance, and MY also was the main input (Figure 2). However, similar to the Animal and Diet models, the inclusion of DMI as a predictive variable increased the r^2^ from 42.36% (Animal + Diet I) to 50.68% (Animal + Diet II). In addition, MY was the main input of Animal + Diet models I (38.28% of total EME variance) and II (38.28% of total EME variance) (Figure 2).

### 3.5. Models Evaluation for Predicting Eenteric Methane Emission

#### 3.5.1. Proposed Models

Animal model I (without DMI) predicted EME with higher mean bias (−3.89 versus −0.72 g/day), lower concordance correlation coefficient (CCC; 0.53 vs. 0.72), r (0.61 vs. 0.74), and bias correction factor (C_b_; 0.87 vs. 0.98), and higher (*p* > 0.05) RMSEP (15.5 ± 0.46 vs. 13.7 ± 0.47% of observed EME) compared to Animal model II (with DMI; Table 7; Figure 3 and Figure 4), indicating that DMI improved the EME prediction in models with only animal as the input (MY and BW^0.75^). 

Dry matter intake also improved EME prediction in Diet models (Table 7; Figure 3 and Figure 4). Diet model II (with DMI) predicted EME with lower mean bias (−0.34 vs. −14.72 g/day), higher CCC (0.78 vs. 0.08) and r (0.81 vs. 0.28), and lower (*p* ≤ 0.05) RMSEP (10.0 ± 0.81 vs. 14.1 ± 1.18% of observed EME) compared to Diet model I (without DMI; Table 7; Figure 3 and Figure 4). However, Animal + Diet model II (with DMI) predicted EME with lower mean bias (−9.36 vs. −14.99 g/d) but similar CCC (0.79 vs. 0.72), r (0.81 vs. 0.78), and C_b_ (0.98 vs. 0.92) and also similar (*p* > 0.05) RMSEP (9.2 ± 1.03 vs. 10.2 ± 1.17% of observed EME) than Animal + Diet model I (without DMI; Table 7; Figure 3 and Figure 4). The Animal + Diet II, Diet II, and Animal + Diet I models predicted EME with the lowest (*p* ≤ 0.05) RMSEP (9.2 ± 1.03; 10.0 ± 0.81; and 10.2 ± 1.17% observed EME) among the six proposed models (Figure 4). 

#### 3.5.2. Comparison with External Models

The 43 external models evaluated in this study (Appendix A) predicted EME with RMSEP ranging from 12.9 to 52.2% of the observed EME, while our six proposed models predicted with RMSEP ranging from 9.2 to 15.5% of observed EME (Figure 4). Only Nielsen et al.’s (2013) models II and III ([23]; Appendix A) and Storlien et al.’s (2014) model III ([24]; Appendix A) predicted EME with similar (*p* > 0.05) RMSEP (13.1 ± 2.67; 12.9 ± 1.95; and 13.3 ± 2.81% observed EME) compared to that of our proposed models (Figure 4). The IPCC 2006 Tier II model [12] overestimated EME in a mean of 28.8 g/cow/day and with higher RMSEP (22.9 ± 2.09% of observed EME) compared to our proposed models (Figure 4). The IPCC 2019 Tier II model [25] underestimated EME in a mean of 8.8 g/cow/day and also had higher RMSEP (19.7 ± 2.53% of observed EME) than our proposed models (Figure 4).

## 4. Discussion

We propose in this study that EME intensity (g methane/kg MY) may be explained and decomposed into two components: EME yield (g methane/kg DMI) and feed conversion (kg DMI/kg MY). Our first hypothesis that feed conversion has greater impact on EME intensity than EME yield was confirmed. Our findings also confirmed that milk yield improvement reduces EME intensity [6,8]. The effect of dilution of the nutrient requirements for maintenance [9] has been considered the main driver explaining the reduction in EME intensity with animal productivity improvement [6].

We demonstrated here that the feed conversion (the inverse of feed efficiency) was a robust driver which explained the reduction in EME intensity with increased milk yield. Although considerable advances have been made to improve productivity and to reduce EME intensity in the dairy cattle industry [6], international climate agreements have been proposed calling for at least a 30% reduction in global methane emissions from 2020 levels by 2030 [7]. This scenario imposes several challenges as the dairy industry continues to meet the growing global demand for dairy foods and reduce its greenhouse gas emissions. According to FAOSTAT [26], the world produced 746 billion liters of milk from 277 million cows, with an overall mean of 2.7 metric tons milk/dairy cow/year in 2021. Developed regions produced half of the milk but with only 18% of the dairy cows in the world; thus, we estimated that overall productivity was 7.5 and 1.7 metric tons milk/dairy cow/year in developed and developing regions. Therefore, several opportunities yet exist to reduce EME by increasing dairy cattle productivity and feed efficiency, especially in developing countries with large dairy cattle populations and relatively lower productivity compared to developed nations.

Ruminal methane formation is an oxide reduction reaction done by archaea (single-cell micro-organisms lacking a cell nucleus), from hydrogen and carbon dioxide and other carbon sources, and it plays a key role in the thermodynamic balance of rumen microbes [27]. Hydrogen and carbon dioxide are formed mainly from the microbial degradation of carbohydrates, and the methane represents the largest sink of hydrogen in the rumen [2]. However, methane emission also represents a source of gross energy loss (2% to 12%) during animal digestion of feed, and this has been associated mainly with intake level, diet composition, dietary additives such as methane inhibitors, rumen microbial population, and animal host physiology and genetics [1,2,3].

As expected, DMI improved the EME prediction in our models. In the Animal model, MY and BW^0.75^ had positive correlations with EME due to its positive correlation with DMI. In the Diet model, OMD combined with DMI was the main dietary input affecting EME, because of its direct association with the amount of degraded carbohydrate by microbes in the rumen [1,2,3]. An enteric methane emissions reduction with increased diet EE was demonstrated by our models probably due to rumen methanogenesis inhibition with lipids supplementation [1,2]. Fatty acid supplementation can reduce rumen protozoa, which are microbial groups that release high amounts of hydrogen from glucose fermentation and are hosts for methanogenic archaea [28,29]. In addition, the rumen biohydrogenation of polyunsaturated fatty acids may contribute to the reduction in hydrogen release in the rumen of cattle [30].

As our proposed models and external models that were evaluated are empirical (i.e., from experimentation) models, the accuracy and precision are population-dependent. As expected, as our models were derived from wide experimental datasets, the EME prediction was improved in comparison with several external models. Only Nielsen et al.’s (2013) models II and III [23] and Storlien et al.’s (2014) model III [24] (Appendix A) predicted EME with similar adequacy compared to our models. Nielsen et al.’s (2013) models II and III use DMI, fatty acids, and neutral detergent fiber in the diet as inputs, while model III from Storlien et al. (2014) used DMI and fatty acids in the diet to predict methane emissions.

We observed that model complexity with the use of variables related to the animal (MY and BW^0.75^) and diet (OMD and EE) combined with DMI (Animal + Diet II model) improves EME prediction. Other studies also have reported improvement in prediction when the model structure becomes more complex [1,31]. However, as all inputs may not always be measured in commercial dairy farms, the future application of our complex model can be limited. Although Animal models I and II had lower accuracy compared to the most complex model (Animal + Diet model II), it had a suitable performance and it was better than several previously published models, including the IPCC 2006 model Tier II [12] and IPCC 2019 model Tier II [25]. Since most modern dairy farms keep track of both milk yield and body weight of their dairy cattle, we recommend our Animal models as an alternative to our more complex model if all its inputs are not well known. Our proposed EME models can be applied in the national inventory of greenhouse gas emissions [25], environmental analysis at dairy farms levels [5], and to help dairy cattle farmers to identify the best strategies to reduce methane emissions.

## 5. Conclusions

We proposed in this study that EME intensity may be explained by EME yield and feed conversion (the inverse of feed efficiency). We confirmed that feed conversion has a greater impact on EME intensity compared to EME yield. In addition, increasing the milk yield reduces EME intensity due to feed conversion enhancement. Therefore, feed conversion improvement should be prioritized for reducing methane emission in dairy cattle systems.

Our model improved EME prediction compared with 43 external models published in the literature, including the IPCC 2006 Tier II and IPCC 2019 Tier II models. Only the Nielsen et al. (2013) and Storlien et al. (2014) models predicted EME with similar adequacy compared to our models. Our more complex model which used predictive variables related to animal (milk yield and metabolic body weight), diet (organic matter digestibility and ether extract), and dry matter intake improved EME prediction from dairy cows. However, one of the models we developed using just variables related to the animal can also be a viable option to be used for estimating EME if other inputs required for our more complex models are not known.

## Figures and Tables

**Figure 1 animals-13-01857-f001:**
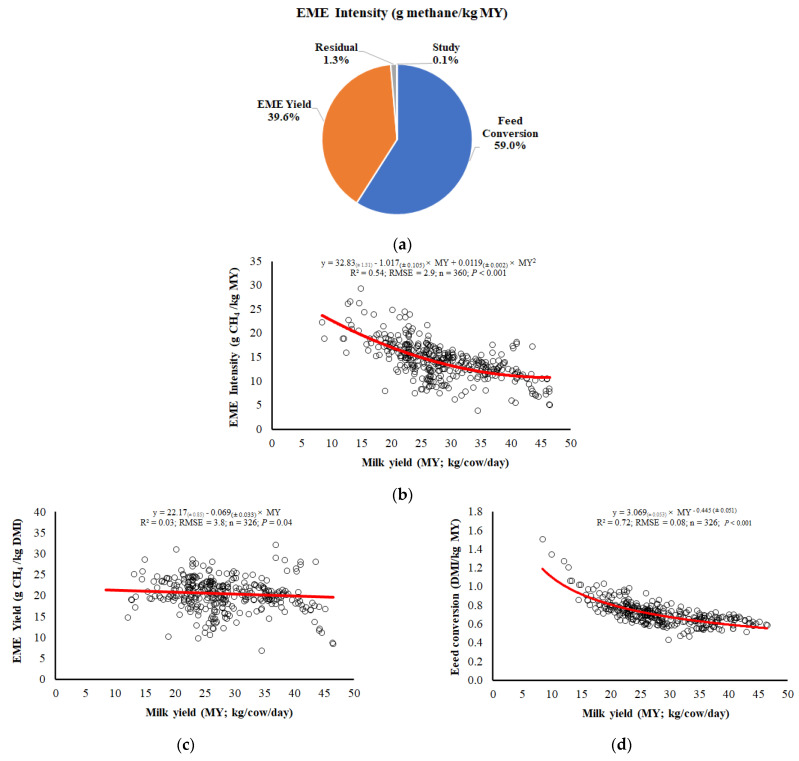
(**a**) Variance component analysis of the effect of enteric methane emission (EME) yield (g methane/kg DMI) and feed conversion (kg dry matter intake (DMI)/kg milk yield (MY)) on EME intensity (g methane/kg MY), where EME intensity = EME yield × feed conversion, and relationship between (**b**) EME intensity and milk yield, (**c**) EME yield and milk yield, and (**d**) feed conversion and milk yield. RMSE = root-mean-square error. *n* = number of treatment means (from Table 1).

**Figure 2 animals-13-01857-f002:**
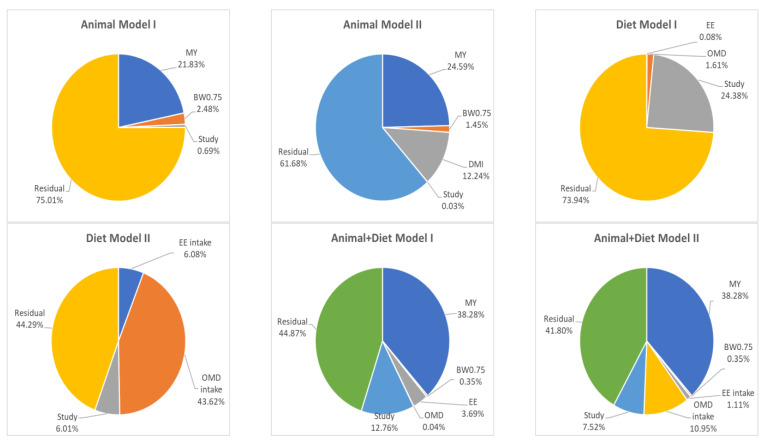
Variance component analysis (% total variance) of the proposed predictive models for enteric methane emission of lactating dairy cows (Table 6). BW^0.75^ = metabolic weight. DMI = dry matter intake. EE = extract ether. MY = milk yield. OMD = digestible organic matter intake.

**Figure 3 animals-13-01857-f003:**
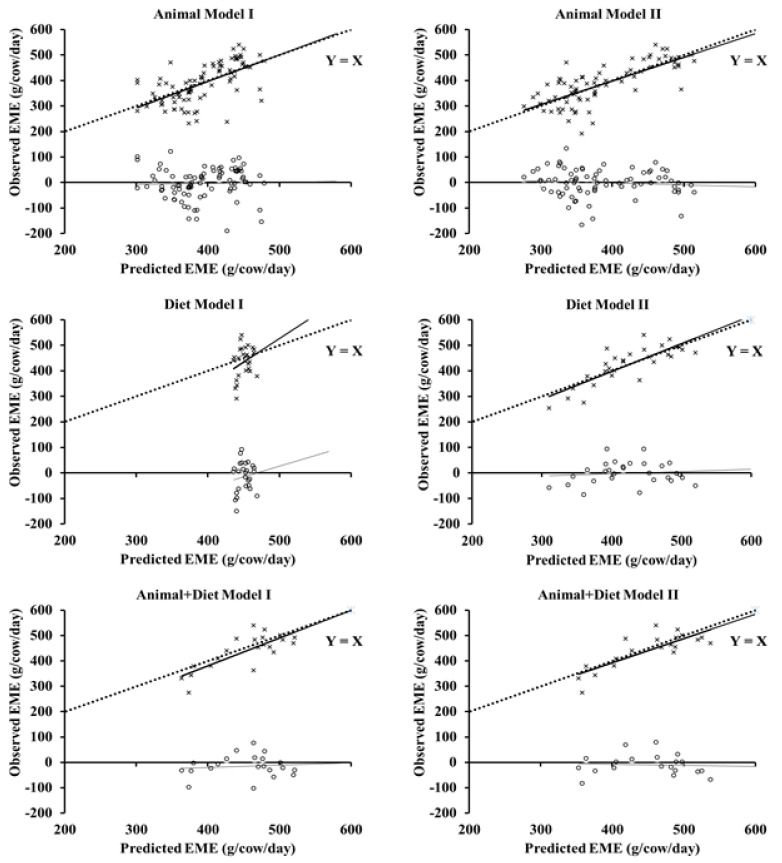
Relationship between observed (squares) and residual (observed−predicted; circles) enteric methane emission (EME) with predicted EME for lactating dairy cows using the proposed models (Table 6).

**Figure 4 animals-13-01857-f004:**
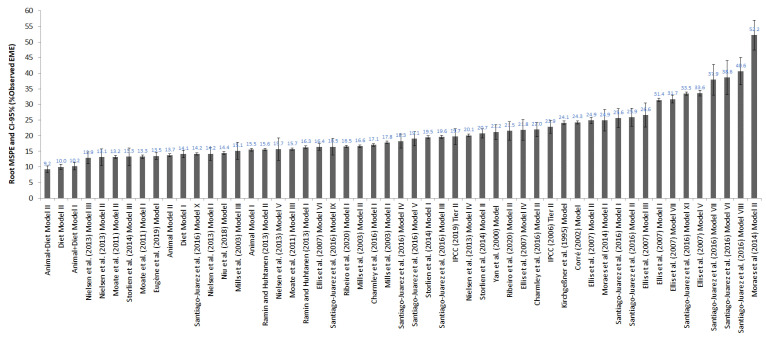
Values of root-mean-square prediction error (MSPE) and its confidence interval (CI 95%) of the enteric methane emission (EME) predictive models for lactating dairy cows. The proposed model is shown in Table 6, and 43 external models are described in Appendix A.

**Table 1 animals-13-01857-t001:** Descriptive statistics of the complete dataset used for developing and evaluating the predictive equation of enteric methane emission (EME; g/cow/day) of lactating dairy cows.

Item ^1^	Mean	Median	Maximum	Minimum	SD ^2^	*n* ^3^
**Animal**						
Body weight (BW), kg	603.80	605.00	714.00	409.00	65.00	362
Milk yield (MY), kg/day	28.06	27.00	46.50	4.90	8.10	384
4% FCM, kg/day	31.78	32.30	42.60	16.90	5.67	71
Days in milk	119	104	302	31	59	360
Milk protein, g/kg	32.51	32.50	39.80	26.10	2.43	308
Milk fat, g/kg	39.94	40.20	53.80	25.90	5.04	322
Milk lactose, g/kg	47.00	46.80	52.70	40.10	2.05	268
Dry matter (DM) intake, kg/day	19.39	18.80	28.50	9.10	3.68	360
**Diet**						
Forage in diet, g/kg DM	594.05	599.50	856.70	283.00	108.77	292
DM, g/kg	475.23	461.00	849.00	134.00	141.37	169
Organic matter (OM), g/kg DM	925.19	927.00	990.00	887.00	13.65	267
NDF, g/kg DM	355.00	348.00	603.00	37.00	61.26	292
Crude protein, g/kg DM	166.97	164.50	251.40	78.00	18.55	300
Ether extract, g/kg DM	39.53	36.00	84.00	19.70	14.31	198
Starch, g/kg DM	203.09	219.50	382.00	5.00	73.12	188
NFC, g/kg DM	412.85	416.00	486.00	261.00	42.41	65
DM digestibility, g/kg	692.44	694.00	798.00	492.00	46.80	151
OM digestibility, g/kg	718.95	713.00	836.00	520.00	50.07	177
NE_L_, MJ/kg DM	6.74	6.70	8.10	5.86	0.41	88
**Enteric Methane Emission**						
Production, g/cow/day	385.65	386.46	748.80	36.00	98.51	419
Yield, g/kg DMI	20.30	20.19	40.00	1.45	4.46	363
Intensity, g/kg MY	14.76	14.30	39.96	1.00	4.98	385

^1^ 4% FCM = fat-corrected milk (kg/d); NDF = neutral detergent fiber (g/kg DM); NFC = non-fiber carbohydrates (g/kg DM); NE_L_ = energy net of lactation (MJ/kg DM). ^2^ Standard deviation. ^3^ Dataset of treatment means was built after reviewing 125 experiments from 115 peer-reviewed papers (Appendix A; the complete dataset in Excel^®^ file is available in an open research data repository, Donadia and Oliveira, 2023 [16]).

**Table 2 animals-13-01857-t002:** Descriptive statistics of the dataset used for developing predictive equations of enteric methane emission (EME; g/cow/day) of lactating dairy cows.

Item ^1^	Mean	Median	Maximum	Minimum	SD ^2^	*n* ^3^
**Animal**						
Body weight (BW), kg	603.80	605.00	714.00	409.00	63.40	270
Milk yield (MY), kg/day	27.92	27.00	46.50	4.90	7.89	273
4% FCM, kg/day	31.29	31.80	42.60	16.90	6.02	50
Days in milk	121	106	302	31	58	261
Milk protein, g/kg	32.66	32.50	39.80	26.10	2.44	216
Milk fat, g/kg	40.39	40.55	53.80	25.90	4.88	226
Milk lactose, g/kg	47.20	47.20	52.70	42.60	2.00	184
Dry matter (DM) intake, kg/day	19.27	18.80	28.20	9.10	3.32	255
**Diet**						
Forage in diet, g/kg DM	600.17	600.00	856.70	283.00	114.21	215
DM, g/kg	477.85	468.50	894.00	134.00	149.75	134
Organic matter (OM), g/kg DM	924.40	926.00	990.00	887.00	13.85	190
NDF, g/kg DM	357.10	351.00	603.00	37.00	66.64	209
Crude protein, g/kg DM	166.37	164.00	251.40	78.00	19.77	217
Ether extract, g/kg DM	37.16	33.00	84.00	19.70	13.37	133
Starch, g/kg DM	205.56	221.00	382.00	22.00	72.11	141
NFC, g/kg DM	415.11	420.00	486.00	261.00	49.00	35
DM digestibility, g/kg	697.32	696.50	798.00	492.00	47.40	118
OM digestibility, g/kg	725.47	717.35	836.00	520.00	51.00	140
NE_L_, MJ/kg DM	6.71	6.74	7.80	5.86	0.53	64
**Enteric Emission Methane**						
Production, g/cow/day	382.16	386.00	656.00	36.00	96.40	301
Yield, g/kg DMI	20.34	20.19	40.00	1.45	4.66	258
Intensity, g/kg MY	14.71	14.32	39.96	1.00	4.84	274

^1^ 4% FCM = fat-corrected milk (kg/d); NDF = neutral detergent fiber (g/kg DM); NFC = non-fiber carbohydrates (g/kg DM); NE_L_ = energy net of lactation (MJ/kg DM). ^2^ Standard deviation. ^3^ Dataset of treatment means was built after reviewing 90 experiments from 80 peer-reviewed papers (Appendix A; the complete dataset in Excel^®^ file is available in an open research data repository, Donadia and Oliveira, 2023 [16]).

**Table 3 animals-13-01857-t003:** Descriptive statistics of the dataset used for evaluation of predictive equations of enteric methane emission (EME; g/cow/day) of lactating dairy cows.

Item ^1^	Mean	Median	Maximum	Minimum	SD ^2^	*n* ^3^
**Animal**						
Body weight (BW), kg	603.70	602.00	712.00	479.00	70.0	92
Milk yield (MY), kg/day	28.39	27.00	45.20	8.40	8.63	111
4% FCM, kg/day	32.95	33.80	42.40	23.90	4.65	21
Days in milk	114	104	247	38	59	99
Milk protein, g/kg	32.17	32.25	36.80	27.30	2.38	92
Milk fat, g/kg	38.86	38.55	52.40	26.70	5.28	96
Milk lactose, g/kg	46.57	46.20	52.00	40.10	2.11	84
Dry matter (DM) intake, kg/day	19.70	18.50	28.50	12.65	4.43	105
**Diet**						
Forage in diet, g/kg DM	576.95	571.00	800.00	361.50	90.38	77
DM, g/kg	465.23	448.00	666.00	145.00	104.41	35
Organic matter (OM), g/kg DM	927.14	930.00	955.00	900.00	13.03	77
NDF, g/kg DM	349.70	344.00	441.00	273.00	44.86	83
Crude protein, g/kg DM	168.54	166.00	236.00	145.00	14.87	83
Ether extract, g/kg DM	44.39	42.00	80.00	21.00	15.04	65
Starch, g/kg DM	195.68	208.00	300.00	5.00	76.39	47
NFC, g/kg DM	410.20	414.50	474.00	321.00	33.77	30
DM digestibility, g/kg	675.00	670.00	745.00	584.00	40.63	33
OM digestibility, g/kg	694.27	691.00	760.00	620.00	37.71	37
NE_L_, MJ/kg DM	6.79	6.51	8.10	6.28	0.57	24
**Enteric Methane Emission**						
Production, g/cow/day	394.58	392.32	748.80	190.86	103.59	118
Yield, g/kg DMI	20.20	20.19	32.83	7.85	3.94	105
Intensity, g/kg MY	14.97	14.30	35.48	5.48	5.33	111

^1^ 4% FCM = fat-corrected milk (kg/d); NDF = neutral detergent fiber (g/kg DM); NFC = non-fiber carbohydrates (g/kg DM); NE_L_ = energy net of lactation (MJ/kg DM). ^2^ Standard deviation. ^3^ Dataset of treatment means was built after reviewing 35 experiments from 35 peer-reviewed papers (Appendix A; the complete dataset in Excel^®^ file is available in an open research data repository, Donadia and Oliveira, 2023 [16]).

**Table 4 animals-13-01857-t004:** Comparison of animal performance, diet, and enteric methane emission data from studies with different methods to quantify enteric methane emission in lactating dairy cows.

Item ^1^	Method ^2^				*p*-Value ^3^
	Respiration Chamber	SF6	GreenFeed^®^	Other	
**Animal**					
Metabolic weight (BW^0.75^), kg	128.98 ^a^ ± 1.11 (*n* = 188)	125.00 ^b^ ± 1.11 (*n* = 114)	131.46 ^a^ ± 1.44 (*n* = 52)	134.16 ^a^ ± 3.28 (*n* = 8)	<0.01
Milk yield (MY), kg/day	31.82 ^b^ ± 0.91 (*n* = 192)	29.63 ^c^ ± 0.91 (*n* = 132)	35.46 ^a^ ± 1.23 (*n* = 52)	33.12 ^abc^ ± 2.81 (*n* = 8)	<0.01
Milk protein, g/kg	32.94 ^a^ ± 0.19 (*n* =161)	31.99 ^ab^ ± 0.23 (*n* = 105)	32.03 ^ac^ ± 0.38 (*n* = 40)	34.85 ^ac^ ± 1.69 (*n* = 2)	<0.01
Milk fat, g/kg	39.91 ^ab^ ± 0.58 (*n* = 169)	38.25 ^a^ ± 0.61 (*n* = 111)	38.99 ^ab^ ± 0.85 (*n* = 40)	43.01 ^ab^ ± 3.52 (*n* = 2)	0.04
DMI, kg/day	21.06 ^b^ ± 0.41 (*n* = 191)	20.76 ^b^ ± 0.43 (*n* = 125)	23.42 ^a^ ± 0.62 (*n* = 40)	23.25 ^ab^ ± 1.72 (*n* = 4)	<0.01
**Diet**					
Forage in diet, g/kg DM	596.56 ^a^ ± 12.33 (*n* = 178)	543.79 ^b^ ± 14.92 (*n* = 64)	569.28 ^ab^ ± 17.78 (*n* = 42)	623.83 ^a^ ± 38.94 (*n* = 8)	<0.01
NDF, g/kg DM	346.53 ± 7.32 (*n* = 163)	344.78 ± 8.02 (*n* = 81)	344.11 ± 10.35 (*n* = 42)	397.69 ± 25.40 (*n* = 6)	0.23
Crude protein, g/kg DM	166.70 ± 1.37 (*n* = 182)	165.66 ± 2.25 (*n* = 68)	171.20 ± 2.79 (*n* = 44)	159.00 ± 7.56 (*n* = 6)	0.29
Ether extract, g/kg DM	42.58 ^a^ ± 1.52 (*n* = 134)	36.09 ^b^ ± 2.45 (*n* = 35)	34.24 ^b^ ± 2.90 (*n* = 25)	28.02 ^b^ ± 6.97 (*n* = 4)	<0.01
Starch, g/kg DM	191.96 ± 9.11 (*n* = 113)	204.93 ± 11.97 (*n* = 48)	211.56 ± 16.77 (*n* = 23)	164.38 ± 36.65 (*n* = 4)	0.42
OM digestibility, g/kg	709.81 ^a^ ± 7.57 (*n* = 105)	674.03 ^b^ ± 8.17 (*n* = 44)	691.87 ^ab^ ± 10.70 (*n* = 26)	693.48 ^ab^ ± 32.24 (*n* = 2)	<0.01
**Methane Enteric Emission**					
Production, g/cow/day	445.3 ^a^ ± 10.61 (*n* = 218)	406.0 ^b^ ± 10.75 (*n* = 142)	470.0 ^a^ ± 14.52 (*n* = 53)	458.0 ^a^ ± 38.32 (*n* = 6)	<0.01
SEM, g/cow/day	22.28 ^b^ ± 1.97 (*n* = 207)	27.55 ^a^ ± 2.03 (*n* = 136)	22.32 ^b^ ± 2.92 (*n* = 45)	13.42 ^b^ ± 7.48 (*n* = 6)	0.03
Yield, g/kg DMI	21.20 ± 0.47 (*n* = 191)	20.19 ± 0.50 (*n* = 125)	20.55 ± 0.77 (*n* = 40)	17.89 ± 2.23 (*n* = 4)	0.13
Intensity, g/kg MY	14.53 ± 0.36 (*n* = 192)	15.38 ± 0.44 (*n* = 132)	14.35 ± 0.69 (*n* = 52)	13.53 ± 2.04 (*n* = 6)	0.38

^1^ DMI = dry matter intake; NDF = neutral detergent fiber; OM = organic matter; SEM = standard error of mean. ^2^ Least squares mean ± standard error (*n* = observation). ^3^ F-test to compare studies with different enteric methane emission methods. ^a–c^ Means values in the same row with different superscripts differ (*p* < 0.01).

**Table 5 animals-13-01857-t005:** Pearson correlation coefficients for the relationships among animal and diet variables and enteric methane emission (EME; g/cow/day) from lactating dairy cows in the full dataset ^1^.

Item ^2^	BW^0.75^	MY	4% FCM	DIM	DMI	NDF	ADF	NFC	Starch	EE	CP	OMD
EME	0.358	0.385	0.429	0.019	0.466	−0.010	−0.004	−0.025	−0.078	−0.104	0.086	0.078
BW^0.75^		0.554	0.526	0.195	0.567	−0.134	0.056	−0.022	0.048	0.123	−0.256	−0.320
MY			0.965	−0.295	0.830	−0.407	−0.067	0.520	0.308	0.160	−0.187	−0.416
4% FCM				−0.402	0.841	−0.359	−0.058	0.477	0.222	0.088	−0.168	−0.309
DIM					−0.101	0.235	0.227	−0.206	−0.231	0.050	0.010	−0.006
DMI						−0.344	0.036	0.433	0.207	0.035	−0.154	−0.419
NDF							0.645	−0.828	−0.434	−0.330	−0.104	0.369
ADF								−0.631	−0.614	−0.281	−0.136	0.277
NFC									0.644	0.080	−0.332	0.091
Starch										−0.010	−0.233	−0.402
EE											0.062	−0.397
CP												0.253

^1^ From complete dataset (Table 1). ^2^ BW^0.75^ = metabolic weight (kg); MY = milk yield (kg/d); 4% FCM = fat-corrected milk (kg/d); DIM = days in milk; DMI = dry matter intake (kg/d); NDF = neutral detergent fiber of diet (g/kg DM); ADF = acid detergent fiber of diet (g/kg DM); NFC = non-fiber carbohydrates of diet (g/kg DM); EE = ether extract of diet (g/kg DM); CP = crude protein of diet (g/kg DM); OMD = organic matter digestibility of diet (g/kg).

**Table 6 animals-13-01857-t006:** Proposed models for predicting enteric methane emission (EME; g/cow/day) of lactating dairy cows.

Models	Equations ^1^
Animal I (without DMI)	EME = 123.29 _(±50.89)_ + 3.32 _(±0.76)_ × MY + 1.49 _(±0.51)_ × BW^0.75^; AIC = 2761.4; *n* = 243
Animal II (with DMI)	EME = 87.68_(±61.61)_ + 2.52_(±1.14_) × MY + 0.582_(±0.563)_ × BW^0.75^ + 8.25_(±2.63_) × DMI; AIC = 1830; *n* = 228
Diet I (without DMI)	EME = 550.21_(±131.29)_ − 0.669_(±0.557)_ × EE − 0.094_(±0.173)_ × OMD; AIC = 866.9; n = 79
Diet II (with DMI)	EME = 133.49_(±43.62)_ − 0.025_(±0.02)_ × EE × DMI + 0.021_(±0.003)_ × OMD × DMI; AIC = 778.8; n = 73
Animal + Diet I (without DMI)	EME = −58.23_(±155.05)_ + 5.09_(±1.26)_ × MY + 2.87_(±1.05)_ × BW^0.75^ − 1.49_(±0.50)_ × EE + 0.06_(±0.016)_ × OMD; AIC = 762.5; n = 73
Animal + Diet II (with DMI)	EME = −28.22_(±113.15)_ + 1.74_(±1.79)_ × MY + 1.75_(±1.06)_ × BW^0.75^ − 0.048_(±0.023)_ × EE × DMI + 0.015 _(±0.004)_ × OMD × DMI; AIC = 770.2; n = 73

^1^ MY = milk yield (kg/d); BW^0.75^ = metabolic weight (kg); AIC = Akaike’s information criteria; DMI = dry matter intake (kg/d); EE = ether extract (g/kg DM); OMD = organic matter digestibility (g/kg); *n* = dataset from Table 2.

**Table 7 animals-13-01857-t007:** Summary of statistical measures to assess the adequacy of proposed models using regression between observed (Y) and model predicted (X) enteric methane emission (EME) of lactating dairy cows.

Item	Proposed Model ^1^					
	Animal		Diet		Animal + Diet	
	I	II	I	II	I	II
Observed EME (Y), g/d	390.50	383.75	436.32	424.44	437.65	441.43
Predicted EME (X), g/d	394.39	384.47	451.04	424.81	452.64	450.79
Mean bias (Y − X), g/d	−3.89	−0.72	−14.72	−0.34	−14.99	−9.36
*n* ^2^	85	76	27	29	21	20
Intercept (*β*_0_)	−21.34	27.19	−393.78	−37.38	−57.70	10.61
Slope (*β*_1_)	1.04	0.93	1.84	1.09	1.09	0.96
*p*-value (H_0_, *β*_0_ = zero and *β*_1_ = 1)	0.80	0.75	0.37	0.85	0.29	0.59
(R^2^)	0.38	0.55	0.08	0.66	0.62	0.65
MSEP ^3^, (g/d square)	3664.88	2778.46	3809.82	1808.71	2008.72	1656.60
Root MSEP, g/d	60.54	52.71	59.39	42.53	44.82	40.70
Partition of MSEP, %						
Error due to mean bias	0.41	0.02	5.92	0.01	10.71	5.03
Error due to slope not equal to 1	0.11	0.75	1.62	1.21	1.05	0.38
Random error	99.48	99.23	92.45	98.78	88.24	94.58
CCC ^4^ (0 to 1)	0.53	0.72	0.08	0.78	0.72	0.79
r ^5^ (0 to 1)	0.61	0.74	0.28	0.81	0.78	0.81
C_b_ ^6^ (0 to 1)	0.87	0.98	0.28	0.96	0.92	0.98

^1^ Full equations for models are specified in Table 6. ^2^ Dataset from 35 experiments of 35 peer-reviewed papers (Table 3). Number of observations was different in models since some inputs are not reported in all studies. ^3^ Mean squared error of prediction. ^4^ CCC = concordance correlation coefficient (CCC = r × C_b_). ^5^ r = correlation coefficient estimate (precision). ^6^ C_b_ = bias correction factor (accuracy).

## Data Availability

Data Availability Statements are available in Donadia and Oliveira, 2023 [16].

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
