# Peer review of "Factors Affecting Enteric Emission Methane and Predictive Models for Dairy Cows"

_animals, 2023, doi:10.3390/ani13111857_

Round 1

Reviewer 1 Report

Dear authors,

Your manuscript titled 'Factors Affecting Methane Enteric Emission and Predictive Models for Dairy Cows' submitted to Animal Nutrition at Special Issue focused on 'Nutritional Strategies to Control Enteric Methane Production of Ruminants' is addressing an interesting topic of research for publication at Animals. It is considered suitable for publication after minor review. See below a few comments/suggestions to be applied by the authors to improve its quality before accepting it for publication.

L17-L18 Replace 'dataset to' by 'dataset to:'

L20 Replace 'Milk yield increase' by 'Increase in milk yield'.

L29 Replace 'This study' by 'In this study'.

L32 Replace 'diets' by 'diet'.

L35 Replace 'Milk yield increase' by 'Increase in milk yield'.

L40 Replace 'sustainability' by 'greenhouse gas emissions'.

L43-L44 Replace 'emis-sions' by 'emi-ssions'.

L69-L70 Replace 'devel-oped' by 'deve-loped'.

L107-L108 Replace 'refer-ences' by 'refe-rences'.

L134-L135 Replace 'as-sociated' by 'a-ssociated'.

L165 Replace 'model' by 'models'.

L177 Replace 'of statistical' by 'from statistical'.

L180 Replace 'ef-fects' by 'e-ffects'.

Table 1: Replace '.Item' by 'Item' and 'Methane enteric Emission' by 'Methane Enteric Emission'.

L241 Replace 'energy net lactation' by 'net energy of lactation'.

L242 Replace 'Treatment means' by 'Dataset was built after reviewing'.

Table 2: Replace Replace '.Item' by 'Item' and 'Methane enteric Emission' by 'Methane Enteric Emission'.

Table 2: Verify data for BW (mean, median, maximum and minimum presented here are the same values than those reported in Table 1).

L253 Replace 'energy net lactation' by 'net energy of lactation'.

L254 Replace 'Treatment means' by 'Dataset was built after reviewing'.

Table 3: Replace Replace '.Item' by 'Item' and 'Methane enteric Emission' by 'Methane Enteric Emission'.

L265 Replace 'energy net lactation' by 'net energy of lactation'.

L266 Replace 'Treatment means' by 'Dataset was built after reviewing'.

L290 Replace 'Milk yield increase' by 'Increase in milk yield'.

L397 Replace 'eeternal' by 'external'.

L456 Delete a space before 'ether extract'.

L482 Replace 'treatment means' by 'Dataset'.

L505-L506 Replace ' 'Treatment means' by 'Dataset' and 'treatment means were' by 'cases under study'.

Figure 2: Move to Supplementary files. Mention that in the text.

Figure 4: Move to Supplementary files. Mention that in the text.

L598-599 Replace 'in the demonstrated' by 'demonstrated'.

L599 Replace 'may be explain by well know' by 'probably due to'.

L624 Replace 'mod-els' by 'mo-dels'.

L632 Replace 'milk yield increase' by 'increase in milk yield'.

L642 Replace 'alternative' by 'option to be used for estimating EME'.

L647 Delete 'used' and replace 'in dataset' by 'in the dataset'.

L648-L649 Replace 'emis-sion' by 'emi-ssions'.

L651-L652 Replace 'Ap-pendix' by 'A-ppendix'.

L728-L729 Replace 'produ-tion' by 'produ-ction'.

L734 Replace 'onlline' by 'online'.

Kind regards,

Reviewer.

Author Response

Dear Reviewer 1,

Thank you for your review.

Point 1: Text changes

Response 1: All your suggestions for text changes were done. We should highlight that the word wrapping at the end of each line is automatic and according to the ANMALS Template.

Point 2:  Verify data for BW (mean, median, maximum and minimum presented here are the same values than those reported in Table 1).

Response 2: The data for BW were verified, and it are correct in Tables 1, 2 and 3.

Point 3:  Figure 2: Move to Supplementary files. Mention that in the text. Figure 4: Move to Supplementary files. Mention that in the text.

Response 3: We cordially disagree with this suggestion. These results are primary outcomes and we would like to keep it in the body manuscript.

Kind regards,

André Soares de Oliveira

Correspondent author

Reviewer 2 Report

I revised the manuscript entitled Factors Affecting Methane Enteric Emission and Predictive Models for 2 Dairy Cows. The study permits the development of some models for the prediction of enteric methane emission intensity in dairy cows. The topic is relevant and fits the scope of the journal. The study is well-designed and the results are clearly described.

Some minor comments

Line 165. These six “models”

166. three “groups”

Lines 287-289. You already mention this in the methodology. Please delete it here.

Line 293. “Figure 1c); However”, replace “;” by “.”

Line 347. Replace “wich” by “which”

Line 353: (P < 0.05) and line 356 (p <0.05): please be consistent with the way to write p values.

Line 379. Please define CCC and Cb again, the first

Line 397. “43 eeternal models”, replace by “43 external models”

Lines 399-403. Please check the reference writing in the text following Animals requirements.

Why have you used some references in the result section?

Line 608-613. Please check the reference

Moderate editing of English language.

Author Response

Dear Reviewer 2,

Thank you for your review.

Point 1: Text changes

Response 1: All your suggestions of text changes were done.

Point 2:  Why have you used some references in the result section?

Response 2: We also evaluated external models, and the citation of their references was necessary. Nielsen et al. 2013 and Storlien et al. (2014) models were cited in the results because it were the best external models evaluated in our study.   

Kind regards,

André Soares de Oliveira

Correspondent author

Reviewer 3 Report

Comments to the Author

Emissions of greenhouse gases from ruminant have a substantial influence on climate change. According to the manuscript, “Factors Affecting Methane Enteric Emission and Predictive Models for Dairy Cows” There are a few errors in the document at various points, thus it has to be revised. My suggestions would be as follows:

1. Heading title: Enteric methane emission vs methane enteric emission, could be consideration.

2. The research's introduction section was nearly completely re-concisely explained by thoroughly with the review literature.

3. In the section on materials and methods, I have noticed:

- Why did the author select a data collection that had not been divided down by location, feed utilization, breed, and crossbreed? What is the response of the error on this variation? It could fit as a fixed effect in an analysis using a linear mixed model.

- According to the author's remark of EME intensity, I wondered if this might be explained by variations in feed component and feed utilization, particularly energy intake.

4. Result and discussion section is required fundamental reconsideration.

- Table 6, after mentioning 225 of the chosen papers at the beginning, why did the author indicate number of data = 243 in equation1?

Author Response

Dear Reviewer 3,

Thank you for your review.

Point 1: Heading title: Enteric methane emission vs methane enteric emission, could be consideration.

Response 1: The heading title was changed.  

Point 2: - Why did the author select a data collection that had not been divided down by location, feed utilization, breed, and crossbreed? What is the response of the error on this variation? It could fit as a fixed effect in an analysis using a linear mixed model.

Response 2: 

Our dataset was also classified by location, feed utilization and breed, and it is available in Excel® File in Donadia and Oliveira 2023 [16]. In addition, the distribution by location, feed system and breed were shown in Results (3.1. Dataset).

Our objective was to develop an overall (global) EME predictive model, and not model each location. The effects of variation between studies were accommodated by two approaches: using studies as a random effect in models; and weighting each observation inverse of the SEM of EME. The variation of study effect of the six proposed was shown in Figure 2 (0.03% to 24.38% total variance). The effects of feed system and breed were not possible to be included as a fixed effect in our models because our dataset has a high concentration of Holstein (85.4%) and TMR system (79%). We have added this limitation in the results. However, we must highlight that despite the limitations our models predicted EME with better adequacy than the 42 external models. 

Point 3: According to the author's remark of EME intensity, I wondered if this might be explained by variations in feed component and feed utilization, particularly energy intake

Response 3: Good point. Our objective was to explain the effects of EME yield and feed conversion on EME intensity. We demonstrated and confirmed that feed conversion (inverse of feed efficiency) was the main driver for the reduction in EME intensity. By deductive logic, any nutrition (i.e. energy diet and intake) or non-nutritional (i.e. genetic, management) that improves feed conversion of dairy cows can potentially reduce EME intensity. The effects of energy intake on EME intensity were modeled by Niu et al. (2018; DOI: 10.1111/gcb.14094), and the authors showed that the increase in energy intake reduced EME intensity. However, based on our study, the effects of energy intake on EME intensity might be explained by improvement in feed conversion. Again, our“big picture” is that the improvement in feed conversion should be prioritized for reducing EME intensity in dairy cattle systems.

Point 4: Result and discussion section is required fundamental reconsideration. - Table 6, after mentioning 225 of the chosen papers at the beginning, why did the author indicate number of data = 243 in equation1?.

Responde 4: As described in detail in the material and methods, a total of 225 peer-reviewed papers were initially found for articles using the keywords “dairy cows” and “methane”. After the adoption of inclusion criteria, 115 peer-reviewed papers with 125 experiments and 419 treatment means containing EME were used in the complete dataset (Table 1; S1). In addition, please note that the “n” in Table 6 (and other Tables) refers to treatment means reported in studies, and not studies.

Kind regards,

André Soares de Oliveira

Correspondent author